# Unsupervised Image to Sequence Translation with Canvas-Drawer Networks

## Abstract

Encoding images as a series of high-level constructs, such as brush strokes or discrete shapes, can often be key to both human and machine understanding. In many cases, however, data is only available in pixel form. We present a method for generating images directly in a high-level domain (e.g. brush strokes), without the need for real pairwise data. Specifically, we train a "canvas" network to imitate the mapping of high-level constructs to pixels, followed by a high-level "drawing" network which is optimized *through* this mapping towards solving a desired image recreation or translation task. We successfully discover sequential vector representations of symbols, large sketches, and 3D objects, utilizing only pixel data. We display applications of our method in image segmentation, and present several ablation studies comparing various configurations.

## 1 Introduction

When an architect reviews a design file, his eyes witness raw image data. But in his head, he is quickly converting these color values into the lines and shapes they represent. Recent works in image classification (Krizhevsky et al., 2012b; Simonyan & Zisserman, 2014; Szegedy et al., 2016) have shown that with deep neural networks, computer agents can learn similar recognition behaviors. The difference appears, however, in how humans and computers *produce* images. Neural networks generally output color matrices that fully detail each pixel, leading to artifacts and blurriness which many recent works have attempted to combat (Goodfellow et al., 2014; Karras et al., 2017). Humans, on the other hand, draw and design on higher-level domains such as brush strokes or primitive shapes. In this work, our aim is to develop computer agents that can operate on a similar level, generating sequences of high-level constructs rather than raw pixels.

While a large amount of past work has been done on reproducing sequences given an observation (Vinyals et al., 2015; Xu et al., 2015), these methods rely on a pairwise dataset of real images and their corresponding sequences, which can be expensive to produce. We focus on the case where such a dataset is not available, and real images are represented only as pixels. Instead, we define a correct sequence as one that recreates a desired image when passed through a given rendering program, such a digital painting software or 3D engine. We present an end-to-end system for training high-level, sequence-based drawing networks without paired data, by optimizing through a learned approximation of such rendering programs.

Our contributions are as follows:

- We establish a simple domain-agnostic method for imitating the behavior of a non-differentiable renderer as a differentiable *canvas network*.

- We propose a framework for training sequence-based *drawing networks* for translation between a low-level and high-level domain (e.g. pixels and brush strokes), without the need for a paired dataset. The main novelty is in utilizing the canvas network to calculate a tractable loss function between generated sequences and real pixel images.

- We develop a method of efficiently extending the canvas-drawer framework to cases where the desired output is a much higher resolution (e.g. 512x512 sketches) and consists of hundreds of sequences, through the introduction of sliding and hierarchical architectures.

We validate our method on a wide range of tasks, including translation of Omniglot symbols and large sketches into sequences of bezier curves, segmentation of architectural floorplans into parametrized bounding boxes, and recreating 3D scenes from a series of 2D viewpoints. We qualitatively show that trained drawer networks easily extend to out-of-distribution examples unseen during training, and we conduct a series of ablation studies on variations such as number of drawing steps, overlap in sliding drawer networks, and the inclusion of a two-layer hierarchy.

## 2 RELATED WORK

Previous works have attempted to learn a mapping between images and their corresponding stroke sequences. (Simhon & Dudek, 2004) learns to refine coarse human sketches by training Hidden Markov Models. (Graves, 2013) presents a general scheme for producing sequences with recurrent neural networks, and (Ha & Eck, 2017) builds on this specifically in the case of producing sketches from a latent space. These methods all depend on a dataset of sketches represented in vector form, whereas our work deals with the case where such a dataset is not available.

Other works have approached the sketch generation problem from a reinforcement learning perspective. (Xie et al., 2013) models a drawing agent to manipulate a digital brush and recreate oriential paintings. (Ganin et al., 2018) considers an end-to-end reinforcement learning agent, and proposes the use of a discriminator as a more accurate reward function. In general, reinforcement learning methods can be unstable and often depend on large amounts of training samples. Our work details a supervised approach where we directly calculate a gradient estimate by modeling a non-differentiable drawing program as a differentiable canvas network.

## 3 FORMULATION

### 3.1 PROBLEM STATEMENT

In our problem, we have a set of pixel images $X$, and we wish to produce the corresponding sequence of high-level constructs $Y$. We will refer to this sequence $Y$ as a sequence of actions. While we do not have a dataset of $X$ and $Y$ pairs, we can infer that the correct $Y$ for a given $X$ is one that recreates $X$ when passed through a renderer $R(Y)$. We assume we have access to such a program $R(Y)$, but it is non-differentiable. Instead, we train a canvas network $C(Y)$ such that $C(y) \approx R(y)$ for $y \sim Y$. We can than train a drawer network $D(X)$ to minimize the pixel distance between $C(D(X))$ and $X$.

Note that the above description is for image recreation. In translation tasks, our data instead consists of $X, X'$ pairs where $X$ is a given hint image and $X'$ is a target image we would like to produce as a sequence of actions. Our objective is then to minimize the pixel distance between $C(D(X))$ and $X'$.

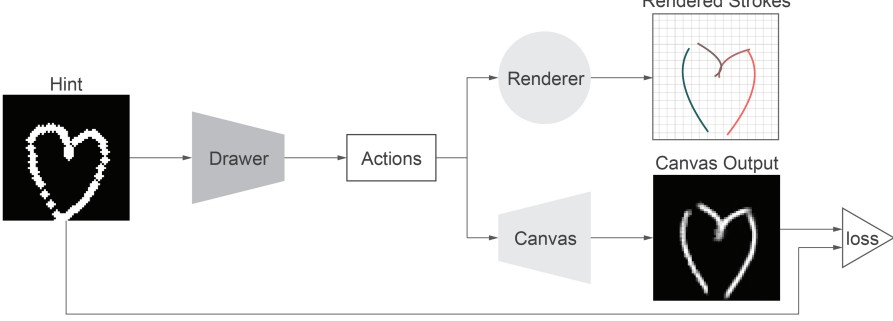

Figure 1: Overall canvas-drawer architecture.

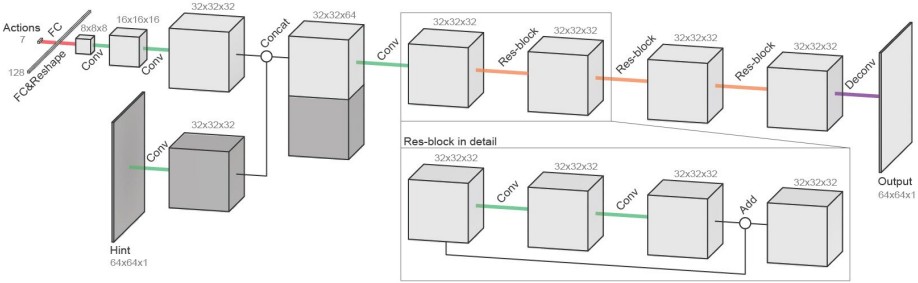

Figure 2: Canvas network architecture and training setup.

## 3.2 CANVAS

In the proposed framework, we view the renderer as a non-differentiable program that maps from a state $x_n$ and action $y_n$ to the next state $x_{n+1}$. We assume there is some starting state $x_0$, and that states are Markovian and contain all past information. We also assume we know the distribution of possible actions. As gradients cannot be passed through the rendering program, we train a differentiable canvas network to act as an approximator by imitating the rendering program for any possible state-action pair. We achieve this by sampling rollouts of $(x_n, y_n, x_{n+1})$ from the renderer, selecting actions $y_n$ randomly and periodically resetting $x_n$ to the initial state $x_0$. We then train our canvas network to recreate $x_{n+1}$, given $x_n$ and $y_n$.

## 3.3 DRAWER

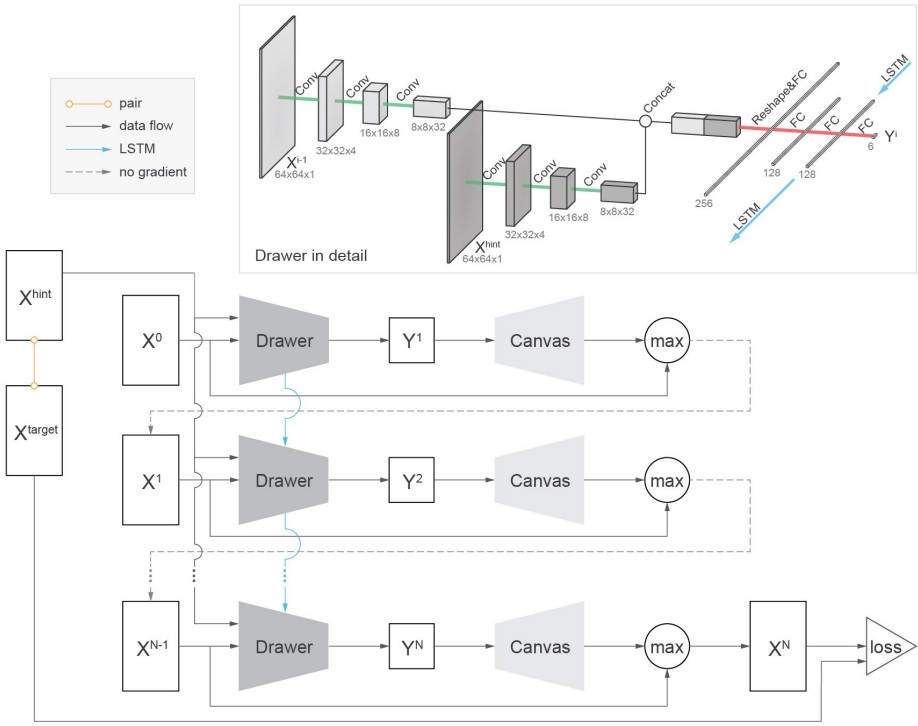

Figure 3: Drawer network expanded for n=3 timesteps.

We define the drawer as a conditional recurrent network that runs for $N$ timesteps. At every timestep $n$, the drawer $D(X, x_n)$ witnesses original image $X$ and the state $x_n$, and selects an action $y_{n+1}$. This state-action pair is given to the renderer $R(x, y)$, which then outputs the next state $x_{n+1}$ which

is given back to the drawer. The true objective of the drawer is to minimize the pixelwise distance between the final state of the renderer $R_{final}(x_{N-1}, y_N)$ and the target image $X'$. However, as $R(x, y)$ is not differentiable, we instead approximate it with canvas network $C(x, y)$. As each action $y_{n+1}$ and state $x_{n+1}$ depends only on the hint image and previous state, the entire graph $C_{final}(x_{N-1}, y_N)$ can be viewed as a function of $X$ and $x_0$.

When training the drawer, we sample $X, X'$ pairs from a dataset of pixel images and optimize towards minimizing pixel distance between $C_{final}(x_{N-1}, y_N)$ and $X'$. $x_0$ is fixed and is generally a blank image. It is important that we train only the parameters of the drawer network, as the canvas network must be frozen to retain its imitation of the rendering program.

In practice, we augment the image inputs with a coordinate layer (Liu et al., 2018), and utilize an LSTM layer (Hochreiter & Schmidhuber, 1997) to capture time-dependencies in the drawer. To assist gradient flow and increase imitation accuracy, we found it useful to define $x_{n+1}$ as a pixelwise maximum between $C(x_n, y_{n+1})$ and $x_n$.

### 3.4 SLIDING

For many real-world cases, image data is large and contains many complex structures. Naive recurrent networks become intractable to train past a few dozen timesteps, and higher resolution images require networks with an increasingly larger amount of parameters.

To mitigate this issue, we extend our method with the idea of a sliding drawer network. Specifically in the case of images, convolutional kernels (Krizhevsky et al., 2012a) have proven to be useful as features in pixel space are generally translation invariant. We take this notion one step further, and assume that *drawing behaviors* are also translation invariant.

We follow this logic and structure a drawing network that slides across the X and Y axes of an image. For every small section of the larger image, our drawer network predicts a short sequence of actions to perform. To prevent artifacts along the borders of sections, we found it useful to slide our drawing network such that overlap occured within neighboring sections. More details on the sliding architecture, as well as the hierarchical variant, are provided in the experiments section below.

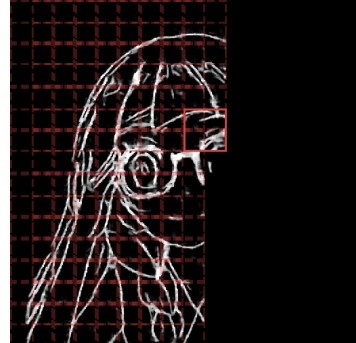

Figure 4: A sliding drawer network slides its receptive field across small sections of a larger image, producing action sequences for each section independently.

## 4 EXPERIMENTS

In the following section, we validate our claims through a range of qualitative results and quantitative ablation studies. It is important to note while pixelwise distance is a helpful directional benchmark, our true goal is to produce accurate high-level actions (e.g. brush strokes, rectangles), which are evaluated qualitatively.

### 4.1 CAN CANVAS-DRAWER PAIRS BE USED TO VECTORIZE SYMBOLS IN AN UNSUPERVISED MANNER?

In a motivating example, we attempt to recreate MNIST digits (LeCun, 1998) and Omniglot symbols (Lake et al., 2015) in a sequence of four bezier curves, akin to physical brush strokes. Each high-level action represents a single curve, defined by two endpoints and a control point. Our state consists of a 64x64 black-and-white image matrix. Since our task is to recreate the symbol provided, both the hint

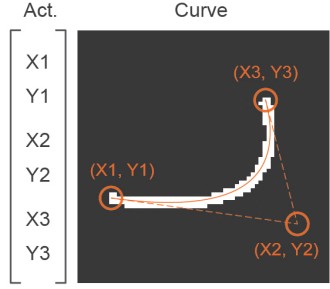

Figure 5: Structure of a single bezier curve.

image and the target image are the same.

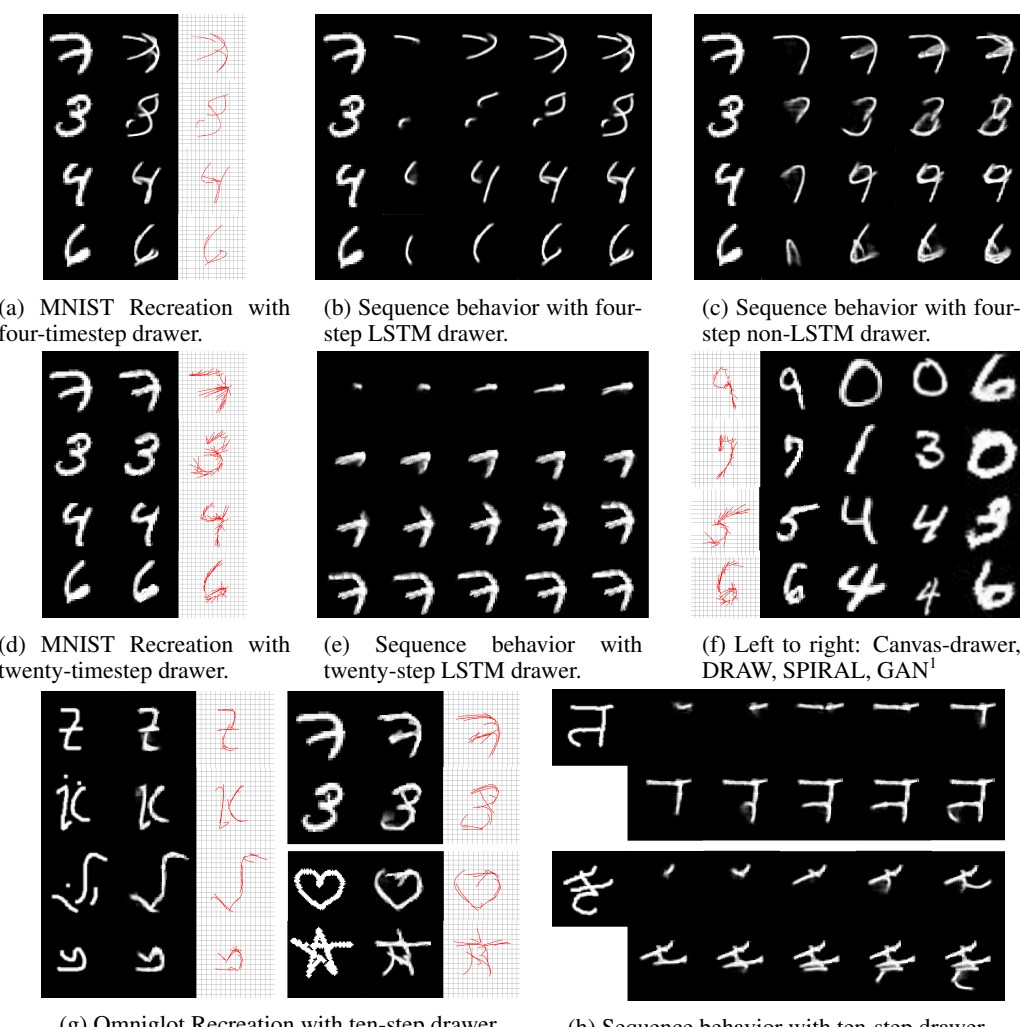

(a) MNIST Recreation with four-timestep drawer.

(b) Sequence behavior with four-step LSTM drawer.

(c) Sequence behavior with four-step non-LSTM drawer.

(d) MNIST Recreation with twenty-timestep drawer.

(e) Sequence behavior with twenty-step LSTM drawer.

(f) Left to right: Canvas-drawer, DRAW, SPIRAL, GAN[1]

(g) Omniglot Recreation with ten-step drawer.

(h) Sequence behavior with ten-step drawer.

Figure 6: MNIST and Omniglot symbol recreation with bezier curve drawing networks. Black-background images refer to pixel outputs, while white-background images showcase parametrized bezier curves rendered in red.

Table 1: MNIST drawer comparison

| Network Structure | Average Pixelwise Loss (L2) |
| --- | --- |
| Canvas-Drawer, Four timesteps, LSTM layer | 0.0175 |
| Canvas-Drawer, Four timesteps, no LSTM layer | 0.0198 |
| Canvas-Drawer, Twenty timesteps, LSTM layer | 0.0080 |
| RL Agent (PPO), Four timesteps | 0.0873 |

A drawer networks lasting four timesteps can accurately produce sequential bezier curves representing distinct segments of the MNIST digits. While non-LSTM drawers achieve roughly equal pixelwise loss, LSTM drawers are more consistent in segmenting digits into concrete curves. In a drawer lasting twenty timesteps, there is a noticeable increase in pixelwise performance, however the curves produced are bended, akin to sketching with tiny strokes rather than smooth motions. We

---

[1]Gregor et al. (2015); Ganin et al. (2018); He (2016)

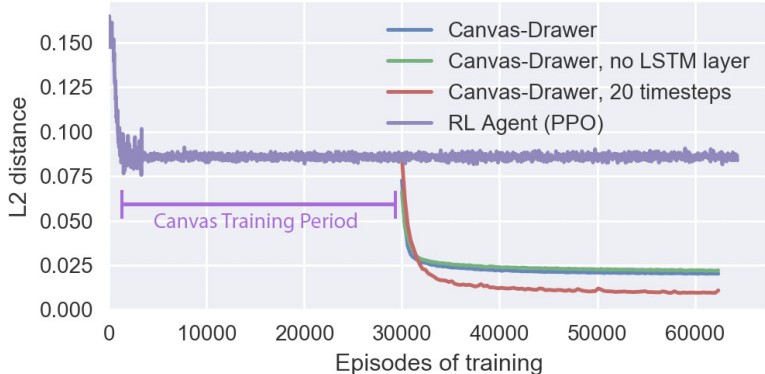

Figure 7: Training curves of various Canvas-Drawer settings. Canvas-Drawer regularly outperforms a typical RL agent in sample efficiency and accuracy, even when accounting for the training time of the canvas network.

Table 2: Sketch Recreation drawer comparison

| Network Structure | Average Pixelwise Loss (L2) |
|---|---|
| Ten timestep sliding drawer, LSTM layer | 0.0530 |
| Four timestep sliding drawer, LSTM layer | 0.0569 |
| Ten timestep sliding drawer, no LSTM layer | 0.0810 |
| Ten timestep sliding drawer, LSTM layer, no overlapping sections | 0.0737 |
| Ten timestep sliding drawer, LSTM layer, fixed curve thickness | 0.137 |

predict this is due to a mismatch in curve thickness, and in later experiments we consider bezier curves with variable thickness to address this issue.

When trained on recreating Omniglot symbols, the drawer network learns a robust mapping that is capable of also recreating symbols that were out of distribution, such as MNIST digits or hand-drawn examples (Figure 6g, right column).

## 4.2 WHAT TECHNIQUES ARE REQUIRED TO SCALE THE CANVAS-DRAWER FRAMEWORK TO LARGE, COMPLEX IMAGES?

Our motivating problem in this section is to recreate a set of 512x512 black-and-white sketches as a sequence of bezier curves. The dataset of sketches was obtained by searching the image host Safebooru for tags "1girl", "solo", and "white background". These images were then passed through a trained SketchKeras (Illyasveil, 2017) network to produce black-and-white line sketches. This recreation problem is challenging as an order of hundreds of curves are required to accurately produce the high-detail sketches.

We made use of a sliding drawer network with a receptive field of 64x64. This drawer is slid across the 512x512 image 32 pixels at a time, resulting in $15 * 15$ overall passes. In each section, our drawer network predicts 10 bezier curves, which are defined as two endpoints, a control point, and a line thickness parameter. To stabilize training, our drawer network skips any small section containing less than 2% white pixels.

The ten-timestep drawer slightly outperforms the four-timestep drawer, in terms of pixelwise loss. However, the curves produced by the four-timestep drawer appear more continuous and follow a smoother outline (Figure 8).

A drawer network without an LSTM layer is still able to reproduce general outlines of the sketch, but misses out on the finer details. Running the drawer with a flood-fill sliding order rather than an increasing X,Y order resulted in nearly identical outputs, which leads us to believe the sliding

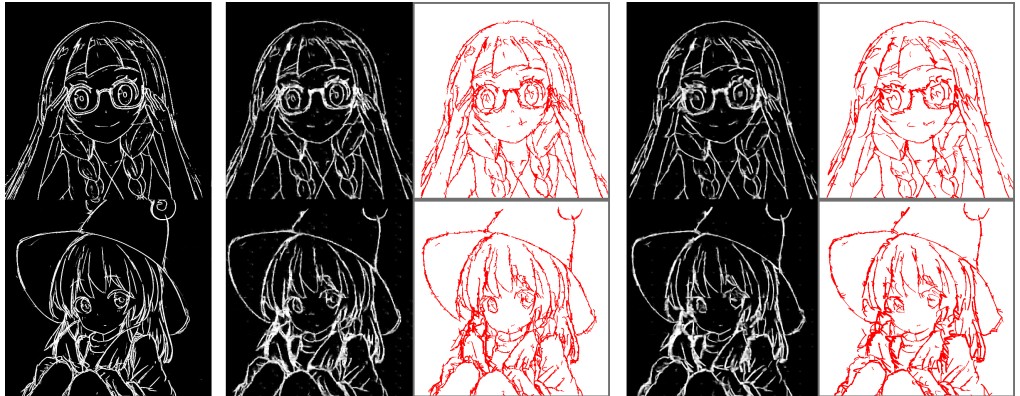

(a) Left: Ground truth original sketches. Middle: Recreation using ten timestep drawer. Left: Recreation using four timestep drawer.

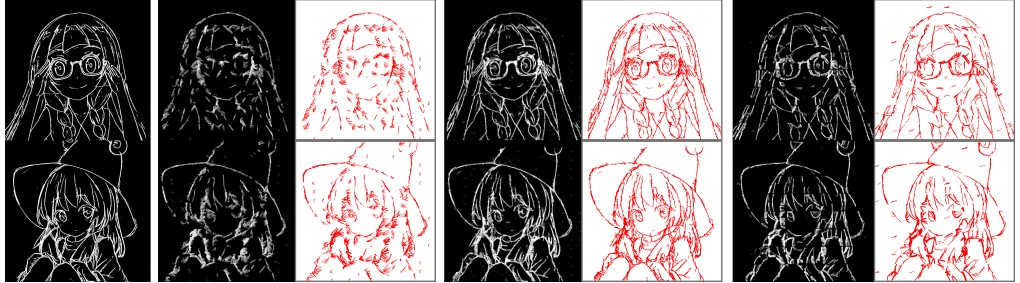

(b) From left to right: Ground truth original sketches. Recreation with non-LSTM drawer. Recreation with flood-fill sliding behavior. Recreation without overlap in sections.

Figure 8: High-resolution sketch recreation using sliding drawer. Black backgrounds represent pixel output, and white backgrounds showcase produced bezier curves in red. Examples from AO (2018); Kushidama Minaka (2017)

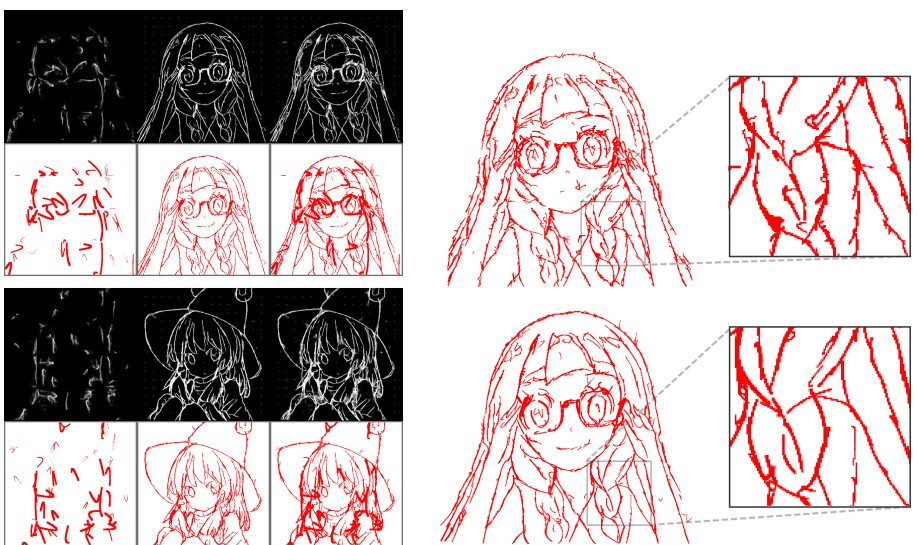

(a) Hierarchical drawer results. Left: 128x128 drawing pass. Middle: 64x64 drawing pass. Right: Final images.

(b) Magnified comparison between non-hierarchical drawer (top) and small hierarchical module (bottom).

Figure 9: Hierarchical drawers learn smoother, more connected drawing behaviors than non-hierarchical variants.

drawer network is generally order-agnostic. Additionally, sliding drawing networks that do not overlap neighboring sections produce curves that fail to intersect properly, leaving artifacts in a grid pattern throughout the produced images.

We also consider hierarchical sliding drawers, in which a large drawing module first performs a pass in 128x128 sections, followed by a small module operating in 64x64 sections. Qualitatively, the curves produced by the larger hierarchical module are messy and inconsistent (Figure 9. However, the small hierarchical module displays an extremely smooth drawing behavior. We believe the larger module may act as a form of regularizer, as it fills in dense areas and allows the smaller module to focus on details.

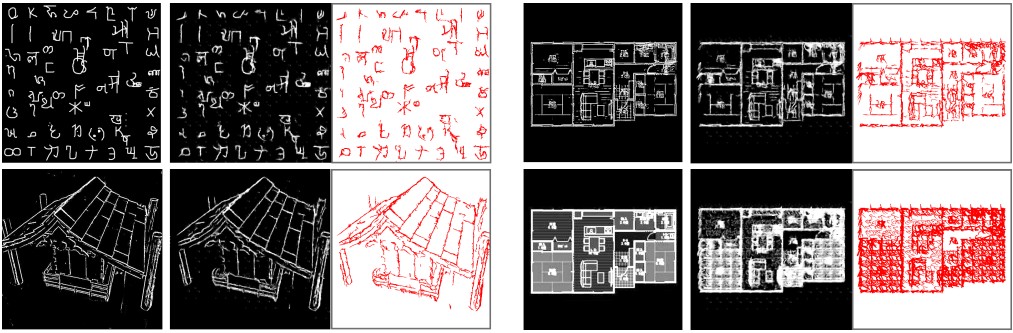

Figure 10: Out-of-distribution test cases on a drawer network trained on the sketches dataset. Bottom-right: The drawer performs different styles of "hatching" to achieve various degrees of shading, similar to traditional pencil sketching techniques.

### 4.3 CAN HIGH-LEVEL DRAWER NETWORKS PERFORM IMAGE TRANSLATION IN COLOR SPACE, AND HOW CAN THIS BE USED AS A FORM OF SEGMENTATION?

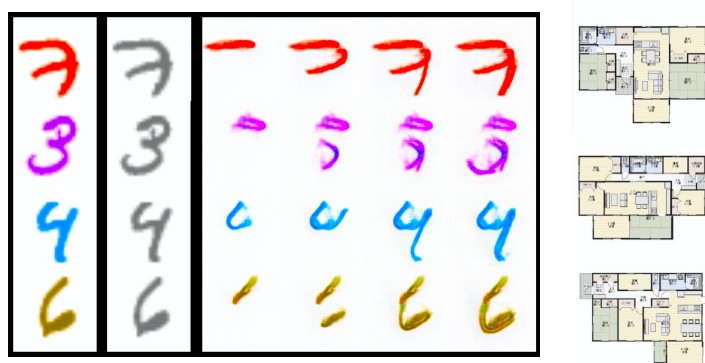

(a) Left: Ground truth. Middle: Pixel hint of a grayscale MNIST digit. Right: Drawer network output. of colored bezier curves.

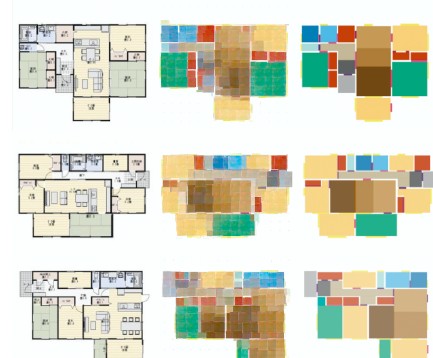

(b) Left: Pixel hint. Middle: Drawer network output of rectangles. Right: Ground truth.

Figure 11: Canvas-drawer pairs are able to translate between grayscale MNIST pixels and colored bezier curves, and between architectural floor-plans and bounding-box room segmentations.

In this section, we examine translation problems where the hint image and target image are different. In Colored MNIST, we train a canvas network on bezier curves with a LAB color component. While the hint image is a grayscale rendering of an MNIST digit, the target image is a colored version with a distinct color for each digit type. To solve this task, a drawer network must not only understand the shape of digits to reproduce them, but also to classify the digits in order to produce the right colored curves.

For the floorplan task, we create a dataset of architectural floorplans (Madori Databank, 2018) and their corresponding room-type segmentations, both in pixel form. The high-level actions are to place rectangles parametrized by two corner points and an LAB color value. By performing this translation in a high-level rectangle space rather than pixel space, we can easily interpret the produced rectangles to form discrete, bounding-box segmentations of various room types. This format is commonly used in design programs and is easily interpretable in comparison to a pixel rendering.

## 4.4 HOW CAN CANVAS-DRAWER PAIRS BE APPLIED TO 3D DOMAINS?

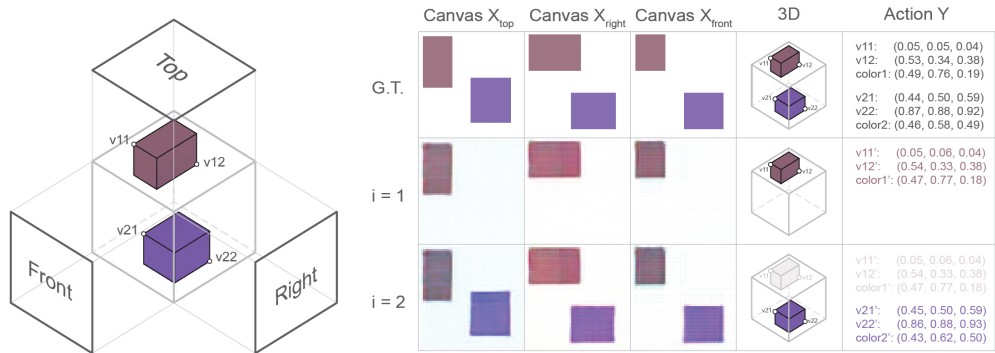

Figure 12: Drawer networks can recreate rectangular prisms in 3D space from a set of 2D observations, without paired data.

Finally, we consider an experiment where high-level actions correspond to corners of a rectangular prism in 3D space. We project these prisms into a set of three orthogonal viewpoints in 2D. Without any paired training data, our drawer is able to accurately reproduce the coordinates and colors of rectangular prisms. This result shows that canvas-drawer pairs can accurately learn behaviors in higher-dimensional space, and future extensions in modeling 3D domains could eliminate the need for expensive paired datasets.

## 5 DISCUSSION

In this work, we presented a general framework for learning low-level to high-level translations without a paired dataset. We approximate the behavior of a rendering program as a differentiable canvas network, and use this network to train a high-level drawer network on recreating symbols and high-resolution sketches, along with translating in colored and 3D domains.

A comparison can be drawn between our method and works in the model-based reinforcement learning and control domain. If the rendering program is viewed as an environment, then the canvas network is similar to a model-based approximation of environment dynamics, and the drawer network is an agent attempting to minimize some image-based cost function.

In addition, our method is similar to an encoder-decoder framework, in the sense that we encode our pixel images into a high-level sequence of actions. However, while most methods learn encoder and decoder networks simultaneously, in our work the decoder is held fixed as we want our encoding space (e.g. action sequence) to be interpretable by humans and arbitrary computer programs.

Finally, a connection can be made with the generative-adversarial network (Goodfellow et al., 2014). An adversarial loss can be seen as a soft constraint for generated images to be close to some desired distribution. This desired distribution often contains useful properties such as resembling natural photographs. In our case, we impose a hard constraint for images to be inside the distribution of a rendering program. Optimizing for recreation through an interpretable constraint is a promising avenue for many unsupervised methods (Zhu et al., 2017; Polyak et al., 2018; Chen et al., 2016).

## 6 LIMITATIONS AND FUTURE WORK

While our method can achieve substantial results on many tasks, we are still far from a universal drawer. The sequential drawer can only handle fixed length sequences. Hierarchical drawers are unable to take full advantage of the larger drawing module, and sketches are still generally drawn in shorter strokes. In addition, certain rendering functions such as hollow rectangles result in unstable behavior of the drawer networks. Due to the differentiable nature of our method, non-continuous action spaces and off-policy exploration strategies would require algorithmic modifications to support.

We believe our work is a stepping stone in achieving unsupervised translation through learned models, and future work could include architectures such as dynamic sequence length or a parameterized receptive field (Gregor et al., 2015). In addition, adjustments like discriminator-based loss or an iteratively updating canvas network would likely improve image quality. If well-behaved parameterizations for complex domains such as 3D modeling, music, or program synthesis are employed, similar canvas-drawer frameworks to could be used to generate interpretable creations while eliminating the roadblock of expensive pairwise data.

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

## 7 APPENDIX

### 7.1 DETAILED EXPERIMENTAL SETUP

The canvas network is defined as a set of residual layers as shown in 2. Every convolutional and feed-forward layer is followed by a rectified linear activation, with a leak factor of $0.2$. During training, we sample pairs of images consisting of a state, an action, and the corresponding next state from a painting program. We train our canvas network with Adam and a step size of $0.001$. No batch normalization layers are utilized. As our example images are constrained between pixel values of $[0, 1]$, we do not use any form of regularization. In the experiments shown, we train our canvas network for a total of 30,000 updates, each with a batch size of 64.

The drawer network similarly contains rectified linear activations and no batch norm layers or regularization, and is trained with Adam and a step size of $0.001$. In addition, we make use of a penalty of $max(0, (y - 0.5)^2 - 0.25) * 30$ on the generated action vector, to discourage the drawer network from producing actions outside the $[0, 1]$ domain. We initialize the final layer of the drawer with a bias of $0.5$, so that actions begin in the center of the canvas.

In the color-based tasks, taking the pixel-wise maximum between the current and next state does not work, as newer strokes should take precedence over older ones. Instead, we parametrize our canvas network to produce a tensor of $[\text{width}, \text{height}, \text{LAB}+\alpha]$. The final dimension contains three color channels, as well as an alpha channel to indicate transparency. When training the drawer, rather than taking a pixel-wise maximum, we instead take a pixel-wise weighted average of $x_n * (1 - \alpha) + x_{n+1} * \alpha$.

In all reported experiments, results are shown on a test set that is separate from the training data.

