# OpenReview forum: "Unsupervised Image to Sequence Translation with Canvas-Drawer Networks"
_ICLR.cc/2019/Conference_

### Official Review · AnonReviewer2 · 2018-11-02
**The evaluation is weak to show its usefulness, despite a nice idea in the underexplored subject**

**Rating:** 4
**Confidence:** 4

**Review:**

To generate a sequence of high-level visual elements for recreation or translation of images, the authors propose differentiable "canvas" networks and "drawer" networks based on convolutional neural networks. One of the main ideas is the replacement of the "canvas" networks instead of non-differentiable "renderer" to end-to-end train the whole model with mean-squared error loss. It seems to be a novel approach to optimize drawing actions. It is reasonable to use separate networks to approximate the behavior of renderer and to fix the parameters of the "canvas" networks to maintain the pretrained rendering capability.

Integrating the high-level visual constructs for recreation or translation of images is to eliminate or attenuate visual artifacts and blurriness, as mentioned in the introduction of the paper. Qualitative comparison with the other state-of-the-art methods is shown in Figure 6f; however, it fails to show significant improvement over them. Quantitative results do not include in the comparison, but only for the ablation study to determine the proposing method. Although the paper proposes an interesting approach to enhance an image generation task, the provided evidence is weak to support the argument, which should be useful for their criteria.

Moreover, experimental details fall short to ensure the validity of experiments. How do you split the dataset as train/val/test? Are the reporting figures (L2 loss) from test results? How are the statistics of the datasets you used?

In Related Work, the authors describe "reinforcement learning methods can be unstable and often depend on large amounts of training samples." Many RL methods use various techniques to stabilize the learning, and this argument alone cannot be the grounding that the supervised approach is better than RL. Unsupervised learning also needs a large amount of data. What is the point of this paragraph (the second paragraph in Related Work)?


Quality:
  Figure 1-3 are taking too much space, which might lead to exceeding 8 pages.

Clarity:
  The experimental procedure is not clear. Please clarify the issues mentioned above. It is not hinder to understand the content; however, the writing can be improved by proof-reading and correcting a few grammatical errors.

Originality and significance:
  Using the differentiable "canvas" networks to avoid non-differentiable "renderer" is a novel approach as far as I know.

Pros:
  Differentiable drawing networks are underexplored in our community.

Cons:
  It failed to show the excellency over pixel-wise generation methods and limited to simple visual elements, line drawings or box generations. This work does not explore "brush strokes" in paintings.


Minor comments:

- In Related Work, the inline citation should be "Simhon & Dudek (2004)" instead of "(Simhon & Dudek, 2004)", and this may apply to the others.

- In Figure 2, the Hint should be x_n, the current state, or target image X for regeneration (X' for translation)?

- In 4.1, a typo, "Out state consists of" to "Our state consists of".

---

> ### Author Response · Authors · 2018-11-19
> **New experiments + Clarification on target comparisons**
>
> Hi, thanks for the response and comments! Some quick clarifications and fixes:
>
> “Integrating the high-level visual constructs for recreation or translation of images is to eliminate or attenuate visual artifacts and blurriness …. It failed to show the excellency over pixel-wise generation methods”:
> The key aspect of our work is to operate in a space where pixel-based representations fail. In real world use cases, such as engineering, design, and art: pixels are not the general representation that professionals use, and it is hard to edit the results in pixel format. In these fields, 2D and 3D objects are represented as high-level representations, for example, vector-based 2D engineering drawings and 3D solid modeling. While we can compare pixel-wise performance and visual artifacts as a useful metric, recreation of images as sequences is an inherently different task than a pixel-based recreation.
>
> “Quantitative results do not include in the comparison”: We’ve conducted additional experiments, directly comparing with an off-the-shelf reinforcement learning algorithm (PPO), showcasing our improvements in terms of accuracy and training time.
>
> “experimental details fall short to ensure the validity of experiments”: We’ve addressed this in the latest revision, with a section in the Appendix containing the detailed experimental details.
>
> “In Related Work, the authors describe "reinforcement learning methods can be unstable and often depend on large amounts of training samples”: We understand that such a claim can seem ungrounded, and we’ve addressed this in the latest revision by conducting experiments comparing our method vs. a typical RL algorithm on the MNIST example -- see Fig 7.
>
> While our method has clear limitations, we believe our contributions are significant as this work is a step into the field of self-learned image-to-sequence translation, which remains relatively unexplored.

---

> > ### Comment · AnonReviewer2 · 2018-11-30
> > **Question about evaluation method**
> >
> > 1. You only mentioned training/test sets in Appendix 7.1. How do you find the hyper-parameters you used?
> > 2. In Table 1, how about to compare with pixel-based generative methods? And, could you report standard deviations to see the significance?
> > 3. Do you believe that `Average Pixelwise Loss` alone is sufficient to compare with the other models?

---

### Official Review · AnonReviewer1 · 2018-11-02
**Simple and working idea, insufficient evaluations**

**Rating:** 6
**Confidence:** 5

**Review:**

This is an interesting paper with simple idea and good results. I like the fact that the authors adopt simple autoencoder-like models instead of GANs or RL.
Following are a couple questions that I am concerned about:
1. Is the ordering of strokes important at all? I suspect that a drawer model that outputs 10 strokes in one pass could perform the same. It might be unnecessary to learn an RNN in this context. Can the authors comment on this?
2. Quantitative evaluations are not well-presented. In Table 1 and Table 2, it is better to normalize pixel wise loss so that the readers could understand the actual error on each pixel.
3. Section 4.3 and 4.4 do not have any quantitative evaluations.
4. How does this system compare with other works, like GANs or RL? Quantitative comparisons are preferred.

The limitation of the proposed approach is also clear: first it is limited to one kind of curves (like a primitive shape in graphics); second it does not learn when to stop, which is already mentioned in the discussion.

---

> ### Author Response · Authors · 2018-11-19
> **New experiments + Clarifications**
>
> Hi, thanks for the comments! We have conducted additional experiments and address some concerns below:
>
> “Is the ordering of strokes important at all … might be unnecessary to learn an RNN”: This point is discussed in the MNIST/Omniglot and Sketch experimental sections of the paper. We show qualitative and quantitative comparisons of an LSTM network and a non-LSTM network, and the LSTM network outperforms the non-LSTM on a regular basis.
>
> “Quantitative evaluations are not well-presented. In Table 1 and Table 2, it is better to normalize pixel wise loss”: This is a good design change, and we have updated the tables to show normalized L2 loss. In addition, we have performed a new comparison to an off-the-box RL method and show the training curves in a graph on Figure 7.
>
> “Section 4.3 and 4.4 do not have any quantitative evaluations”: In this research, our ablation studies and comparisons are conducted mainly on the MNIST/Omniglot and Sketch experiments, as they are more traditional experimental settings. 4.3 and 4.4 are new problem domains that we developed for this paper, and we can best measure how well our method performs through a qualitative rather than quantitative comparison, as these settings are unconsidered in previous work.

---

### Official Review · AnonReviewer3 · 2018-11-02

**Rating:** 4
**Confidence:** 4

**Review:**

This paper presents an unsupervised method for generating images in a high-level domain (brush strokes and geometric primitives). The proposed system is comprised of two neural networks: the drawer D and a forward model C of an external renderer R. The latter is trained on the rollouts produced by sending random actions to R. The forward model is then freezed and used to train D, i.e., the network that repeatedly interacts (sends commands) with the C to produce a desired image. Since everything is differentiable, D can be optimized via regular gradient descent.

Pros:
+ The paper is written clearly and relatively easy to read
+ The idea to replace the non-differentiable renderer with a differentiable approximation makes sense. Pure RL setups (i.e., in [Ganin et al., 2018]) are quite sample inefficient and hard to train due to high variance of REINFORCE.
+ The proposed method is tested both in 2D (drawings and floor plans) and 3D (prisms) domains and yields relatively good results.

Cons:
- The datasets used in the paper are quite simplistic. I’m wondering how hard it is to train a forward model for more complex data. At the very least, I would want to see how the method handles the 3D experiment when the view is not axis-aligned and there is more variety in the number of primitives and their types.
- The performance of the method especially on drawings and floor plans is not excellent. The drawer tends to reproduce line art with small disjoint strokes - very different from how humans would accomplish this task. A similar observation can be made for floor plans (the system outputs small pixel-like boxes that tile bigger rooms). This suggests that recovered commands do not really correspond to the higher-level structure of the input. Unlike in RL approaches, injection of prior knowledge about the data (e.g., the floor plan should be reproduced using the minimum possible number of rectangles) seems problematic within the proposed framework.
- It’s unclear how to use the approach for non-continuous actions (e.g., choosing types of primitives in 3D or different instruments in music).
- It seems the method may suffer from significant exploration problems in more complex settings. Consider an image of a rectangle that the system should reproduce. Say, it initially outputs a box that doesn’t intersect with the target one. The gradient of l^2-distance between those two images in the pixel space is non-zero but it is zero w.r.t. the action taken since no small change of the action parameters would make the generated box intersect with the target (assuming that the target is far from the model’s output) so l^2 will stay the same.
- I would love to see some comparison (preferably quantitative - speed of training, quality of reconstructions, etc.) to an RL system. So far, in the paper, there is only one figure showing a couple of images produced by such system.

Notes/questions:
* Section 2, paragraph 2: The systems by Xie et al. and Ganin et al. are very distinct. The former models the appearance of a single stroke while the latter is more similar to the present paper and synthesizes the entire image using strokes of a predefined appearance.
* Section 3.3, paragraph 3: “pixel-wise maximum” - it seems to be a fairly restrictive setup which only works when the model increases intensity of pixels.
* Section 3.4: This is a straightforward idea and is not novel (e.g., already used in some demonstrations of the method in [Ganin et al., 2018])
* Section 4.2, paragraph 2: During training, do you use all the patches or randomly sample them? Is your loss computed per patch or for the entire image?

In summary, the paper presents an interesting idea but the execution needs some improvement (especially, in terms of evaluation) before the paper is ready to be accepted to the conference.

After going through the authors' comments and the revised version of the paper, I keep the rating as is. The paper needs a more convincing evaluation section as well as some clean up (e.g., references to figures and tables in the text)

---

> ### Author Response · Authors · 2018-11-19
> **New experiments + Addressed points**
>
> Hi, many thanks for the detailed review! We have conducted additional experiments, and address some points below:
>
> “The datasets used in the paper are quite simplistic”: Regarding the 3D experiment, it is true that the experiment is simplistic, however that result is mostly a proof-of-concept that differentiable canvas methods can be extended to higher dimensions if setup correctly. The problem of unsupervised translation through a simulator is still relatively unexplored, and compared to current state-of-the-art methods like SPIRAL that operate over MNIST and Omniglot, we go further and show previously unseen results on detailed sketches.
>
> “The performance of the method especially on drawings … reproduce line art with small disjoint strokes”. This is a fair point, and it is an issue that is inherent to the problem at hand, which we mention in the Limitations section.. Since we optimize for pixelwise accuracy, the drawer network prefers to use small, accurate strokes rather than a few long strokes with higher potential for error. We address this issue in our paper via the introduction of a hierarchical network, and qualitatively the smoothness improves (Figure 9). Potential future improvements to this include utilizing GAN-type loss to encourage natural looking strokes, which we suggest in the Discussion.
>
> “It’s unclear how to use the approach for non-continuous actions”: This is another fair point, and we have revised our Limitations section to include it. Since discrete actions are non-differentiable by nature, integrating them into the canvas-drawer setup would require a non-trivial amount of modifications, which could be considered in future work.
>
> “It seems the method may suffer from significant exploration problems”. It is true that exploration is an inherent issue in optimizing through a simulator, as we see in many RL contexts, and the example you give certainly has a possibility of occurring. In our experiments, however, we found that a proper initialization of the actions can mitigate this issue, as we are able to learn a reasonable drawing policy in the floor-plan rectangle experiment. The Con compared to RL methods is that certain off-policy RL methods can utilize hand-engineered exploration policies, whereas we are limited to actions close to the output of our network -- we have updated our Limitations section to include this statement.
>
> “I would love to see some comparison … to an RL system”. We have run additional experiments, with the same experimental conditions, except using an off-the-box RL system (PPO) to produce the actions. We quantitatively compare pixelwise performance and training times -- Figure 7 contains this updated graph.
>
> Notes/Questions
> Section 2, paragraph 2: We have cleared up the distinction between these two methods.
> Section 3.3: It is true that the pixel-wise maximum only works when strokes add to the intensity, and we actually encountered this limitation in our colored experiments. To fix the issue, we introduce an alpha layer in the canvas network, and compute the next state via a weighted average -- we have added details in the Appendix regarding this setup.
> Section 3.4: Our desired point in this section was that our method is extendable to high-resolution images via position independence, a technique not seen yet in the context of unsupervised image-to-sequence translation. We have updated the section to clarify this.
> Section 4.2, paragraph 2: We use all patches, but losses are computed per batch. We have revised the experimental details in the Appendix to mention this.
>
> While there are certainly inherent limitations to our method, we believe our contributions are significant as the idea of a differentiable renderer in the context of sequence generation is relatively unexplored. We have conducted additional quantitative experiments, and show that our method outperforms alternative (RL) in terms of accuracy and sample efficiency.

---

> > ### Comment · AnonReviewer3 · 2018-11-26
> > **Comments**
> >
> > 1. "compared to current state-of-the-art methods like SPIRAL that operate over MNIST and Omniglot, we go further and show previously unseen results on detailed sketches"
> > The SPIRAL paper does contain an experiment in 3D. Moreover, it's unclear if the sketches considered in this work are any more challenging than Omniglot (the network is still operating on small patches which resemble Omniglot symbols)
> >
> > 2. The comparison to an RL method looks suspicious. The baseline doesn't seem to have learned anything reasonable (judging by the plot). A fair comparison should include a successful run of an RL system (and we know such systems exist).
> >
> > On a more general note, it seems that some tables and figures are never referred to in the text so it's hard to understand where they belong in the narrative.

---

> > > ### Author Response · Authors · 2018-11-27
> > > **Response**
> > >
> > > Thanks for the comments. It is true that SPIRAL presents a 3D experiment and we recognize this point. Still, we stress the 2D work in our results are more complex than Omniglot symbols -- in order to create a smooth final sketch, agents need to handle both strokes that leave the section boundaries, and handle overlap between the differing sections. Without these factors, the large sketches that are produced look disjoint and messy (Fig 8b). We also present a hierarchical setup specifically for the case of creating smoother large sketches.
> > >
> > > The RL method did successfully learn, as seen by its initial decrease in L2 loss, however even with hyperparameter tweaks could not improve performance past a local minimum. Other works have shown RL methods approaching stronger performance, but required significantly larger amounts of training episodes. To compare fairly, we allow the RL agent the same amount of training episodes as the differentiable agents we are comparing against.
> > >
> > > Regarding figures, we have added a couple of references when figures are on differing pages than their contexts.

---

### Public Comment · (anonymous) · 2018-11-14
**Results reproduction**

Thanks a lot for the nice work!
Found the paper very interesting and promising for tasks I have in hands and therefore was considering to reproduce the results on the dataset given in the paper, as well as my own dataset. However, didn't find in the paper enough data to reproduce it, including detailed network structure, regularizations, optimizers used to train.
It would be very helpful for reproducing the results sharing the implementation of any of configurations, specified in the paper.

---

> ### Author Response · Authors · 2018-11-19
> **Experimental details added**
>
> Hey, appreciate the interest! We’ve updated the paper with a new revision, including an appendix that goes over the experimental setup in extended detail.

---

### Author Response · Authors · 2018-11-19
**Revision: New experiments and small changes**

We have updated the paper with an experiment comparing our method to a standard RL algorithm (PPO) on the MNIST recreation task. We show improved performance and training speeds, even when taking account the training time of the canvas network. In addition, we have written a more detailed experimental setup in the Appendix, and updated our Limitations section to more clearly state the current boundaries of our method.

---

### Meta-Review · Area_Chair1 · 2018-12-14

**Confidence:** 4
**Recommendation:** Reject

**Metareview:**

This paper was reviewed by three experts. After the author response, R2 and R3 recommend rejecting this paper citing concerns of experimental evaluation and poor quality of the manuscript. All three reviewers continue to have questions for the authors, which the authors have not responded to. The AC finds no basis for accepting this paper in this state.